



# Snow Persistence Explains Stream High Flow and Low Flow Signatures with Differing Relationships by Aridity and Climatic Seasonality

Edward Le[1], Ali A. Ameli[1*], Joseph Janssen[1], John Hammond[2#]

[1] Department of Earth, Ocean and Atmospheric Sciences, The University of British Columbia, Vancouver, British Columbia, Canada

[2] Department of Ecosystem Science and Sustainability, Colorado State University, Fort Collins, CO, United States

[#] Now at U.S. Geological Survey, MD-DE-DC Water Science Center, Baltimore, MD, United States

Correspondence to: Ali A. Ameli (aameli@eoas.ubc.ca)

**Abstract.**

Snow persistence is a globally available metric of snow cover duration that has, thus far, seen limited application to the field of hydrology. This study attempts to explore the controls that snow persistence exerts on streamflow at low and high flow conditions across a diverse spectrum of climatic aridity and seasonality in the United States and Canada. We statistically analyze how snow persistence, aridity, and seasonality conditions interact to control and explain streamflow shape and flashiness at low and high flows. For low flow condition, regardless of climatic aridity and seasonality, a larger snow persistence increases baseflow, reduces low flow variability, and increases the magnitude of extreme low flow relative to average flow. Our results further show that snow persistence becomes a stronger factor in controlling baseflow as well as the magnitude of extreme low flow relative to average flow, in regions with a relatively high aridity and/or with summer-dominant precipitation regimes (or in-phase seasonality). On the other hand, in catchments that are moderately wet to very arid with winter-dominant precipitation regimes (or out-of-phase seasonality), a longer snow persistence could typically lead to a more variability at high flow and a larger magnitude of extreme high flow relative to average flow. This study concludes by demonstrating the relevancy of snow persistence as a globally available streamflow behaviour descriptor and by demonstrating the impacts that climate change may have on snow persistence and ultimately on streamflow behaviour at low and high flows.



## 1. Introduction

Signatures of streamflow hydrograph have long been extracted to analyze catchments' streamflow behavior and explain their internal hydrologic functioning, allowing for the development of generalizable hydrologic knowledge valid for a wide range of divers regions (McMillan, 2021). In practice, streamflow signatures have a wide array of applications in classical hydrologic research, such as process elucidation, environmental decision making, and analyses of land-use or climate change impacts on catchment hydrology (Smakhtin, 2001; Wagener et al., 2007; McMillan, 2020; Sawicz et al., 2011; McMillan, 2021). While there

are many different types of streamflow signatures, shape-based signatures seek to represent the shape and flashiness of hydrographs and they can reveal important insights into baseflow, catchment storage, and anthropogenic impacts on catchment flow paths (McMillan, 2020; Janssen and Ameli, 2021). Changes to global water systems are predicted to occur due to global climate change (IPCC, 2018). Linkages between streamflow signatures and climatic metrics (Addor et al., 2018) can yield important insights on hydrologic functioning in (un)gauged catchments under a changing climate (Hrachowitz et al., 2013; Addor et al., 2018; Yadav et

al., 2007) using globally-available climate observations. However, there is no parsimonious and generalizable process-based framework that can link widely available types of climatic observations to shape-based streamflow signatures, particularly low flow signatures (McMillan, 2020), in snow-dominated catchments (Addor et al., 2018). This paper will attempt to fill a portion of this knowledge gap by examining the extent that a—globally available—snow-related metric can explain various shape-based stream high and low flow signatures across a wide variety of climatic aridities and seasonalities in the United States and Canada.


It has long been known that snow water storage and release has a particularly important role in the behaviour of catchments' hydrology (Linsley et al., 1975; Shaw et al., 2011; Marshall et al., 1994; Dewalle and Rango, 2008). In a human context, snowmelt is at least partially (if not mostly) responsible for providing water for billions of people around the world (Mankin et al., 2015). Previous research has shown that snowpack duration and snowmelt influences the volume and temporal characteristics of water

availability within catchments (Barnett et al., 2005; Beniston and Stoffel, 2014). Snow persistence (SP) is a simple integrated metric of snowpack accumulation and ablation as defined by the fraction of time that snow is present on the ground within a defined period of time, such as during parts of a year (Molotch and Meromy, 2014; Moore et al., 2015) or across an entire year (Hammond et al., 2018b). Hammond et al. (2018a) showed that SP could potentially describe annual water yield, while Eurich et al. (2021) explored links between SP and annual or monthly water yields. Eurich et al. (2021) also found that in combination with other

variables (e.g., topography, geology, climate), the addition of SP in predictive models registered appreciable gains relative to previous modelling techniques – particularly in dry, mountainous regions.

The advantages of using SP as a descriptor of hydrologic behaviour are that (1) it is a theoretically parsimonious predictor of snow's influence on catchment functioning and streamflow response (Hammond et al., 2018a; Kampf and Lefsky, 2016; Eurich et

al., 2021), and (2) it can be easily derived for many regions around the world (Hammond et al., 2018b), enabling the differentiation of areas with permanent, seasonal, or intermittent snow presence. While studies have used alternate snow metrics, such as snowfall fraction of precipitation, to examine linkages with streamflow (Berghuijs et al., 2014a), snow persistence does not require regional fine-scale meteorological data or setting and adjusting temperature and humidity thresholds for the separation of rain and snow to match observations. As continued climate warming reduces seasonal snowpacks and changes snowmelt timing, several remotely

sensed snow metrics, including snow persistence, have been recommended for tracking these changes as well as their hydrologic consequences (Nolin et al., 2021).



Previous studies on the linkages between SP and hydrology have generally focused on the local applicability of SP's descriptive power in a limited range of North American climatic settings. For example, Kampf and Lefsky (2016) used SP to find relationships between SP and peak flow magnitude over time, within Colorado's Front Range. Hammond et al. (2018a) identified relationships between SP and annual streamflow (i.e. annual water yield) in the western U.S. More recently, Eurich et al. (2021) used SP to predict monthly and annual streamflow values in Colorado for ungauged basins. A major advantage of exploring relationships between SP and hydrologic behaviour is that it does not require on-the-ground measurements of variables. Instead, it can be reliably calculated from remote sensing platforms such as the Moderate-Resolution Imaging Spectroradiometer (MODIS) satellite (Hall et al., 2002). This highlights a major research gap wherein data exists on a global scale but has only been used at the local scale, and does not evaluate the general predictive applicability of SP to more diverse climates around the world (Eurich et al., 2021; Kampf and Lefsky, 2016; Hammond et al., 2018a). As an example, North America features a variety of climates ranging from very dry with constant seasonality and no snow (in the southern United States), to moderately wet with high seasonality and significant snowfall in the northern reaches of Canada's Newfoundland and Labrador (Knoben et al., 2018).

Additionally, the previous studies primarily focused on the capability of SP in predicting magnitude-based streamflow signatures such as peak flow magnitude (Kampf and Lefsky, 2016) or mean annual or monthly water yield (Eurich et al., 2021; Hammond et al., 2018a). These streamflow signatures, despite their important implications for water management and security, do not thoroughly explain the internal hydrologic functioning of the catchments or provide linkages between snow behaviour and stream low and high flow variability and flashiness, which are critically important to understand vulnerabilities to hydrologic flood and drought (McMillan, 2020). Insights on both (a) hydrologic functioning and (b) snow's behavioural linkages to the variabilities of stream high and low flows are critically required for the prediction of hydrologic response in ungauged regions, particularly under changing climatic conditions and land uses (Hrachowitz et al., 2013; Evenson et al., 2018).

This study seeks to build a generalizable understanding of the linkages between SP and catchment hydrologic functioning by enlarging the scope of analysis climatically and behaviourally to ultimately evaluate SP's effectiveness as a parsimonious and widely available tool to inform climate change impact analyses, water management, and decision-making, in (un)gauged catchments. In doing so, we adopt a statistical inference method—which acknowledges the interactions amongst catchment climatic descriptors (predictors)—to a hydrologic context by exploring the relationships between SP and shape-based streamflow signatures across a vast gradient of seasonality and aridity in the United States and Canada. Our overarching hypothesis is that there are distinct relationships between SP and shape-based streamflow signatures at different levels of climatic aridities and seasonalities. Specifically, we aim to explore:

1. Can snow persistence explain the shape (flashiness) of streamflow hydrographs at low and high flow conditions?
2. How do aridity and seasonality affect the ability of SP to explaining shape-based signatures at stream low and high flow conditions?

Given that snowpacks are highly vulnerable to global warming (Huning and Aghakouchak, 2018; Shrestha et al., 2021; IPCC, 2021; Siirila-Woodburn et al., 2021), our paper will provide generalizable process-based predictive tools to relate the persistence of snowpacks and streamflow flashiness at high and low flow conditions, for use in (un)gauged catchments, where streamflow data may not exist but SP data are available. These parsimonious predictive tools can then provide water stake holders with a process-based quantitative understanding to better evaluate catchment streamflow responses and make well-informed decisions about water security and environmental disturbances (e.g., flood & drought) under a changing climate.





## 2. Data and Study Sites

There are two major categories of data used in this study, including: 1) climatic data (Sec. 2.1) and 2) streamflow data (Sec. 2.2).
The study period covers the years 2001–2019 and the study sites represent catchments from across North America (Fig. 1).

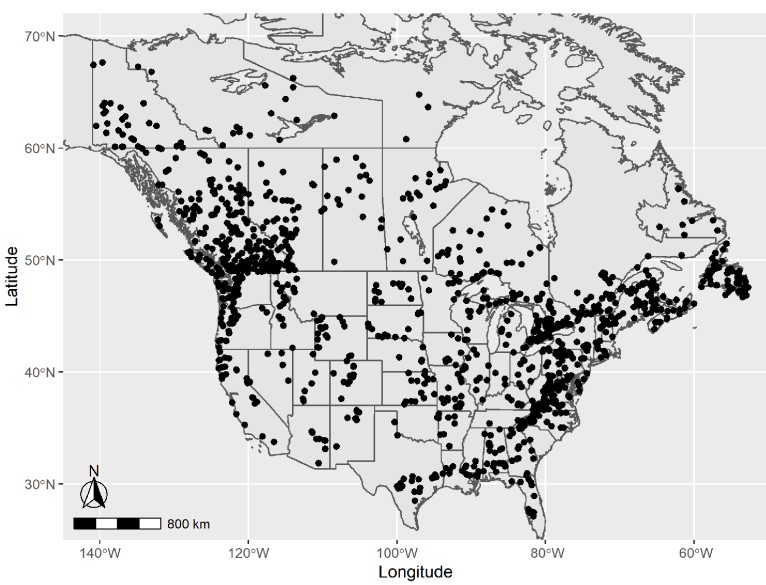

**Figure 1: A spatial map of the 1,187 catchments used in this study.**

### 2.1 Climate Data

We used globally available climatic datasets to increase the generalizability of the research. Catchment-scale long-term average
Snow Persistence (SP), Aridity Index (AI) and Seasonality Index (SI) were calculated for each catchment. Catchment-scale values
of these climatic metrics were calculated using catchment boundary polygons (See Sec. 2.2). Spatial maps and summary statistics
of these metrics are provided in Fig. 2a-c and Table 1. To calculate SP during the study period at each catchment, the 500 m
resolution—MODIS/Terra 8-Day L3 Version 6— snow cover product (MOD10A2) was used (Hammond, 2020b, a; Hall and
Riggs, 2016). This data source represents an iteration of NASA MODIS data that attempts to better correct for previously
discovered issues and features some improved snow cover detection accuracy (Riggs et al., 2016).

Aridity Index is a measure of the relative "dryness" of a given region. It is defined by the ratio between Potential Evapo-
Transpiration (PET) and total precipitation (Budyko, 1974). It can be interpreted by focusing on the critical value of 1. Below 1, a
region can be considered to be wet, whereas, above 1, a region is considered dry (Budyko, 1974). The aridity index has been shown
in previous continental-scale work to be a key factor in classifying hydrologic behaviour (Kuentz et al., 2017). The Seasonality
Index is a dimensionless measure of the synchronicity between precipitation and temperature in a single variable interpreted
between -1 and 1 (Woods, 2009). At the extremes, -1 (out-of-phase) signifies winter-dominant precipitation, 0 signifies uniform
precipitation throughout the year, and +1 (in-phase) signifies summer-dominant precipitation (Berghuijs et al., 2014b). Catchment-
scale long-term average climate aridity and seasonality indices used in this paper were calculated in Janssen and Ameli (2021).
They used daily and monthly temperature and total precipitation data from the ERA5-Land database (Muñoz-Sabater et al., 2021)
as well as monthly PET data released as part of the TerraClimate database (Abatzoglou et al., 2018), to calculate the long-term
average AI and SI values at more than 2,000 catchments (including those catchments used in this study) across North America.



## 2.2. Streamflow Data

A well-distributed set of stream gauges across North America were selected for this study. We started with daily streamflow
observations at 671 and 1,602 stream gauges across the United States and Canada, respectively. These daily streamflow
observations were analyzed to generate long-term streamflow signatures (see Sec. 3.1).

The United States set of stream gauges and their corresponding catchment boundary polygons were obtained from the CAMELS
dataset (Addor et al., 2017). Addor et al. (2017) developed and validated catchment boundary polygons for each gauge; with each
catchment having two sets of area estimates based on two different area estimation methodologies. We then queried the catchments
listed in the CAMELS dataset to get updated data from the  United States Geological Survey (2020). The Canadian gauges were
derived from the HYDAT dataset released by Environment and Climate Change Canada (2021). We derived flow direction and
flow accumulation and ultimately catchments' area and boundary polygon for the Canadian gauges using the Watershed Tool in
ArcGIS. The calculated areas of Canadian catchments were then validated against area information from topographic maps by the
Water Survey of Canada (Environment and Climate Change Canada, 2021). Thus, like the United States catchments, there were
two sets of area estimates available for the Canadian catchments. Catchments used from both countries generally feature minimal
human disruption (e.g., human development, stream diversions, and flow regulations).

## 2.3 Catchment Selection Criteria

From the original sets of catchments, only catchments with the following criteria were selected and used in the statistical analysis:
a)    Catchments with 15 or more years of acceptable streamflow records during the study period. Each year should have fewer
than 10% of days missing streamflow data to be considered as an "acceptable" year.
b)    Catchments with an area difference, between the two alternative approaches, less than or equal to 30% (See Sec. 2.2 for more
details).
c)    Catchments with perennial flow records. Catchments' with a large number of zero flow days were excluded from the analysis.

Following this screening, a total of 1,187 out of 2,273 original catchments were selected. These catchments represent a diverse
variety of North American climatic regions across a wide gradient of snow persistence, aridity, and seasonality (Fig. 2a-c).

## 3. Methods

The methodologies are organized around streamflow signature calculations (Sec. 3.1), Snow Persistence (SP) derivation (Sec. 3.2),
and statistical inference procedures (Sec. 3.3).

## 3.1 Streamflow Signatures

Shape-based streamflow signatures can represent the shape and flashiness of hydrographs to help elucidate relationships in the
baseflow of streams, human impacts on flow pathways, and relationships with storage of water in catchments (McMillan, 2020;
Janssen and Ameli, 2021). In this study, five shape-based streamflow signatures were used to assess the ability of SP to predict
streamflow flashiness at low and high flow conditions. These signatures include the baseflow index, normalized $Q_5$, the slope of
stream low flow duration curve, normalized $Q_{95}$, and the slope of stream high flow duration curve (Table 1).





The Baseflow Index (BFI) describes the proportion of total flows that come from baseflow (flow that is maintained between precipitation events) and is a useful signature in large-sample hydrologic comparisons between catchments (Addor et al., 2017).

BFI was also shown to be connected to the presence of snow through the proxy of snow fraction in a given catchment (Addor et al., 2018). $Q_5$ and $Q_{95}$ were normalized against $Q_{mean}$ to produce normalized $Q_5$ and normalized $Q_{95}$, respectively. Both signatures serve as dimensionless shape-based indices to broadly describe catchments across different regions that are agnostic to the impacts of flow magnitudes (Janssen and Ameli, 2021). The Flow Duration Curve (FDC) aggregates recorded streamflow values against their exceedance probability and is a compact representation of streamflow data (Searcy, 1959). The slope of an FDC curve can

then be taken from the distribution of probabilities of streamflow less than or equal to specified magnitudes to describe the variability of flow within a catchment (Sawicz et al., 2011). Large slopes in the FDC indicate more variable streamflow, whereas smaller slopes indicate damped streamflow variability. We used two slopes (in logarithmic scale) to represent the flow variability, during low flows (Low-FDC; between the 5th and 30th percentiles) and high flows (High-FDC; between the 70th and 95th percentiles).


To estimate long-term values of the signatures for each catchment during the study period, the code and methodology developed in Addor et al. (2017) were used. Details of the mathematical/methodological basis of the code from Addor et al. (2017) were summarized in Table 3 of their study. Note that an $\alpha$ of 0.925 was used to calculate the baseflow index. The spatial maps of the streamflow signatures used in this study are shown in Fig. 2d-i. Table 1 reports some statistics of these signatures.




**Figure 2: The spatial maps of climatic metrics (snow persistence, aridity index, and seasonality index; a-c, respectively) and streamflow signatures (the baseflow index, normalized Q5, Low-FDC, normalized Q95, and High-FDC; d-h, respectively) of the 1,187 catchments used in this study. For all plots except snow persistence, the seasonality index, and the baseflow index, a natural log colour bar transformation was used to emphasize the regional differences between catchments. Note: Geographical scale bars may not be very accurate due to the continental scales of the maps and are just shown for convenience.**






**Table 1. Description of catchment-scale climatic metrics and streamflow signatures. Median and interquartile ranges reflect non-transformed values across the study sites used in this paper. The transformation function for each signature/metric, used in the statistical analysis, is also shown (See Sec 3.3 for more detail). "Ln" denotes natural logarithm, "Sqrt" denotes square-root transformation, "Cbrt" denotes cube-root transformation. The table also describes the statistical model performance and the strength of interaction amongst climatic metrics in explaining streamflow signatures (See Sec 3.3 for more details).**

| Type: | Metric/Signature | Unit | Median (Q1, Q3) | Transformation Function | Linear Model $R^2$ | H-Statistic | |
|---|---|---|---|---|---|---|---|
| | | | | | | AI & SP | SI & SP |
| Climatic Metrics | Snow Persistence (SP) | % Of time snow present between Jan. 1–July 3rd | 47.25 (19.36, 64.58) | - | - | - | - |
| | Aridity Index (AI) | - | 0.7 (0.5, 0.9) | Ln | - | - | - |
| | Seasonality Index (SI) | - | 0.1 (-0.09, 0.3) | - | - | - | - |
| Streamflow Signatures | Baseflow Index (BFI) | - | 0.55 (0.46, 0.64) | - | 0.23 | 0.15 | 0.33 |
| | Normalized $Q_5$ | - | 0.11 (0.05, 0.17) | Cbrt | 0.12 | 0.27 | 0.12 |
| | Low-FDC | - | 3.59 (2.45, 5.57) | Ln | 0.25 | 0.07 | 0.17 |
| | Normalized $Q_{95}$ | - | 3.47 (3.02, 3.99) | Ln | 0.11 | 0.75 | 0.45 |
| | High-FDC | - | 5.29 (4.21, 6.53) | Sqrt | 0.16 | 0.11 | 0.67 |

### 3.2 Snow Persistence (SP)

The SP was determined as the fraction of days of snow presence from January 1-July 3, a period that generally covers peak snow accumulation and complete snow ablation for many parts of North America (Moore et al., 2015). Calculations were performed using MODIS 8-day images (See Sec. 2.1). SP was calculated for the continental United States and the Canada/Alaska region (Hammond, 2020a, b) using methods from Hammond et al. (2018a). Next, SP values were then spatially delineated with catchment boundary polygons (Sec. 2.2) to clip the gridded SP data and were temporally aggregated to calculate catchment-scale averages

during the study period. Spatial maps and summary statistics are provided in Fig. 2a and Table 1, respectively.

### 3.3 Statistical Methods

The multivariate regression analysis was conducted in R version 4.1.0 (R Core Team, 2021) with the R *interactions* package developed by Long (2019) and the R *iml* package by Molnar et al. (2018). Multivariate regression models serve as a classical method to predict streamflow characteristics (Lull and Sopper, 1966; Shu and Ouarda, 2012; Eurich et al., 2021; Janssen et al.,

2021). Here, multivariate linear regression models were developed between each streamflow signature (denoted as response variables, $Y_k$) and climatic metrics including SP ($x_1$), AI ($x_2$), and SI ($x_3$) as model predictors (Eq. 1). The $\beta$s represent regression





coefficients while $\epsilon_k$ represents the error term for the $k^{\text{th}}$ streamflow signature. Interaction terms are included (e.g., $\beta_4 x_1 x_2$; the interaction between SP and AI, $\beta_{5,k} x_2 x_3$; the interactions between AI and SI, $\beta_{6,k} x_1 x_3$; the interaction between SP and SI, or $\beta_{7,k} x_1 x_2 x_3$; the three-way interaction between SP, AI, and SI) to capture the potential interactions amongst climatic metrics.

Indeed, as reflected in Eq. (1), we hypothesized that there are distinct relationships between SP and shape-based streamflow signatures at different levels of climatic aridities and seasonalities.

$$Y_k = \beta_{0,k} + \beta_{1,k} x_1 + \beta_{2,k} x_2 + \beta_{3,k} x_3 + \beta_{4,k} x_1 x_2 + \beta_{5,k} x_2 x_3 + \beta_{6,k} x_1 x_3 + \beta_{7,k} x_1 x_2 x_3 + \epsilon_k \ (1)$$

Variables were transformed to meet the requirements of linear regressions (Pek et al., 2017) and the models were validated using residual, Quantile-Quantile (QQ), and residual histogram plots (See Supplementary Materials; Figs. S1-S5). The developed models were then fed into the *sim_slopes* function from the *interactions* package to visualize the interactions between variables via Johnson-Neyman analysis. This method runs many statistical inference tests for each streamflow signature, so many false positives would normally be expected. To correct for this, the *control.fdr* option is used from the *sim_slopes* function (Esarey and Sumner,

2017; Long, 2019). Johnson-Neyman analysis has been shown as a robust way to analyze the data with an eye for finding numerically-localized regions of statistical significance (Bauer and Curran, 2005). Johnson-Neyman analysis was particularly useful in this study as it allowed us to identify how climatic differences in seasonality and aridity, seen across North America, can have an impact on the predictive power of SP. Finally, to assess the strength of these interactions, a Friedman's *H-statistic* analysis was done with the *Interaction* function from the *iml* package (Molnar et al., 2018). The *H-statistic* is a dimensionless statistic that

assesses the share of the variance that is explained by an interactive effect and is typically interpreted between 0 and 1 (Molnar, 2021). Here, 0 means no interactions exist between the variables, 1 means that all of the variance is explained by the sum of the partial dependence functions (i.e., there is a very strong interaction), and values between 0 and 1 reflect a continuous scale of the strength of the interactions amongst variables (Molnar, 2021). As Molnar (2021) notes, there is some instability possible in the calculation of the *H-statistic* due to the need to estimate marginal distributions. Therefore, we took the average of three random

estimates of *H* values to arrive at a reasonable approximation of the true value.

In summary, the analytical frame of this study is to examine the linkages between streamflow signatures and SP and explore how such linkages vary across a range of aridities and seasonalities. This analysis is supported by multivariate regression models to explore SP's predictive ability across available variable domains, then the primary analysis is carried out with Johnson-Neyman

analysis to identify intervals of statistically significant interactions. Finally, the *H-statistic* is used to quantitatively express the strengths of the interactions.

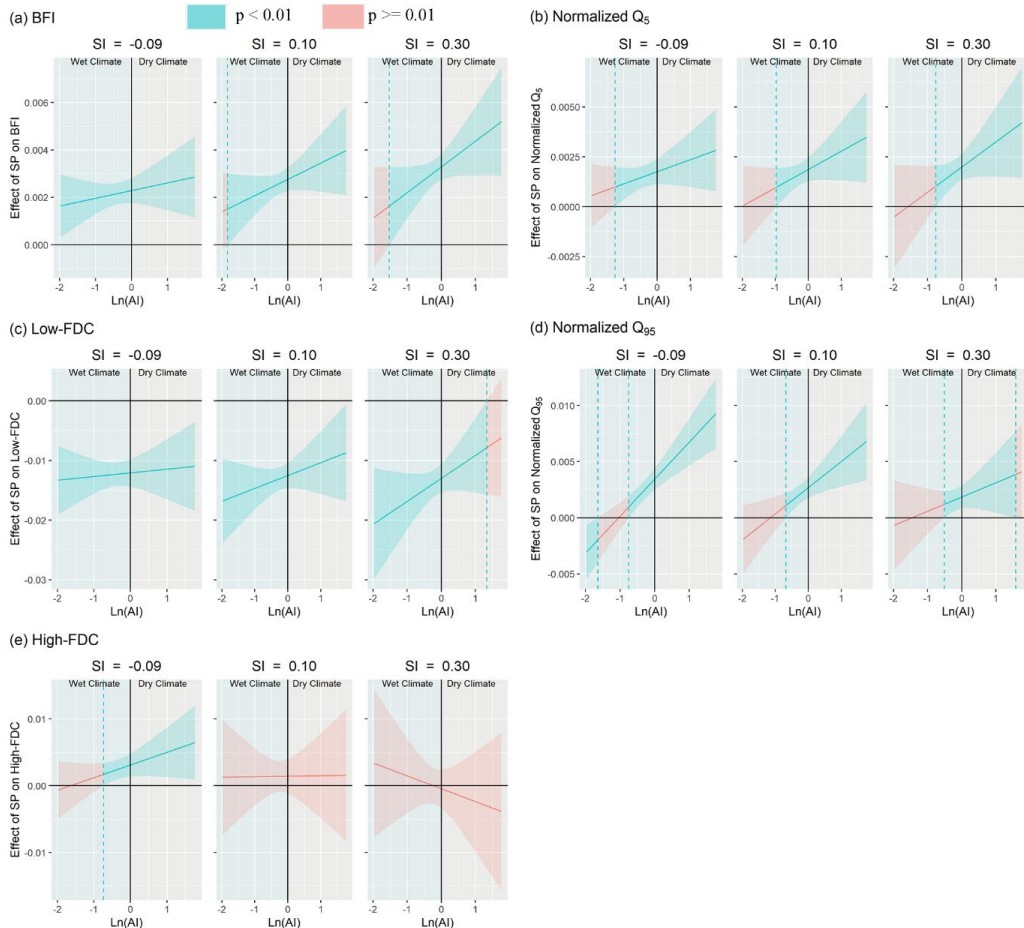

**Figure 3. Effect of Snow Persistence (SP) on shape-based streamflow signatures as a function of aridity index (AI) and seasonality index (SI), using Johnson-Neyman interaction plots. The effects were calculated as continuous functions of both AI and SI (see Eq. 1) but visualized in this figure at three quantiles of SI. Q1 (SI = -0.09) represents out-of-phase seasonality, median (SI = 0.10) represents uniform seasonality, and Q3 (SI = 0.30) represents in-phase seasonality. Blue indicates areas of statistical significance (p < 0.01), while red indicates areas that do not contain statistically significant relationships (p >= 0.01). The y-axes refer to the conditional effect of SP on a given signature at different levels of AI and SI. Required transformations for the variables are listed in Table 1. Diagnostic tests for multivariate regression models including residual, Quantile-Quantile (QQ), and residual histogram plots are shown in Supplementary Materials; Figs. S1-S5.**





## 4. Results

### 4.1 Relationships Between Snow Persistence and Baseflow Index (BFI)

There is a statistically significant positive association between SP and BFI ($p < 0.01$; Fig. 3a), across (almost) the entire spectrum
of aridities and seasonalities used in this study. As aridity increases, the positive association between SP and BFI becomes stronger,
regardless of seasonality level, suggesting a larger influence of snowpack persistence on BFI in dry regions compared to wet
regions. For example, in catchments with very dry climates, e.g., those found in New Mexico and Arizona, there is the strongest
positive association between SP and BFI. The combination of this evidence suggests that SP plays a role in maintaining higher
baseflows in catchments across a range of catchment dryness and seasonality, with the largest impact in dry climates.

### 4.2 Relationships Between Snow Persistence and Normalized $Q_5$

There is a statistically significant positive association between SP and normalized $Q_5$, from moderately wet ($AI > 0.47$) to very dry
catchments, regardless of seasonality conditions ($p < 0.01$; Fig. 3b). This association is notably stronger in drier climates, before
becoming statistically insignificant in extremely wet climates ($AI < 0.47$). Example catchments with such statistically significant
positive associations include catchmnets in Southern California, portions of Alaska, across central North America, and into the
eastern seaboard of the United States. *H-statistic* values quantitatively reflect the importance of the interaction, particularly between
SP and AI, in explaining normalized $Q_5$ (Table 1). The combined evidence depicts, in moderately wet to very dry catchments, SP
increases the relative magnitude of extreme low flows to that of average flows, mitigating fast hydrograph recession, and reducing
the vulnerability to streamflow drought.

### 4.3 Relationships Between Snow Persistence and Low-FDC

Across almost the entire set of studied catchments in both wet and dry climates, and in-phase and out-of-phase seasonalities, there
are statistically significant negative associations between SP and Low-FDC ($p < 0.01$; Fig. 3c). Such associations are an order of
magnitude stronger than the associations between SP and the two other stream low flow shape-based signatures reported above
(BFI, normalized $Q_5$). This suggests that SP could strongly mitigate the variabilities between 5th to 30th percentiles of stream flow
(i.e. mitigating fast recession) and could sustain ample low flows during a stream's driest days.

### 4.4 Relationships Between Snow Persistence and Normalized $Q_{95}$

Generally, in moderately wet to very dry climates, there is a statistically significant positive association between SP and normalized
$Q_{95}$ ($p < 0.01$; Fig. 3d). This association is generally stronger with out-of-phase precipitation seasonality (i.e. winter dominated
precipitation). Such interactive associations are also reflected in the *H-statistic* values (Table 1). Results imply that SP can increase
the ratio of stream high flow to average streamflow, increasing the relative height of "peaks" in catchment hydrographs.

### 4.5 Relationships Between Snow Persistence and High-FDC

Results show a positive association between snow persistence and High-FDC that was only statistically significant in areas that
featured out-of-phase precipitation seasonality with moderately wet to very dry climates (Fig. 3e). This generally includes
catchments in the western United States, with some extension north towards south-central British Columbia (Canada), parts of the
southern United States near northern Alabama, and some catchments in Maritime Canada (e.g., $AI \geq 0.48$, $SI \leq -0.09$). The strong
interactive effects amongst SP and SI are supported by large *H-statistic* values (Table 1). Altogether, this suggests that in



catchments with out-of-phase precipitation regimes and moderately wet to very dry climates, a longer presence of snow on the ground is associated with larger high flow variability or flashier stream high flow patterns.

## 5. Discussion

To create a generalizable understanding of the linkages between snow persistence (SP) and catchment hydrologic behaviour and streamflow response at stream low and high flow conditions, we explored two overarching questions:

1. Can snow persistence describe the shape (flashiness) of catchment hydrologic response?
2. How does climate aridity and seasonality affect the ability of SP to explain shape-based signatures?

The importance of these questions are twofold. First, it helps to evaluate the explanatory capability of a snow-related descriptor of catchment hydrologic behaviour that could be useful for predictions of streamflow signatures in poorly gauged to ungauged catchments. This is particularly powerful given that snow persistence is a remotely sensed and globally available measure of snow cover duration (Hammond et al., 2018b). With the recent development of cloud gap filled versions of daily MODIS (MOD10A1F) and VIIRS (VNP10A1F) snow cover datasets (Hall et al., 2019), and the availability of open-source tools to rapidly process this data (Crumley et al., 2020), calculation of snow metrics for large, unmonitored areas are more feasible than ever before. Second, this study provides a quantitative understanding of the impact of snow pack loss, in the light of climate change, on streamflow characteristics. We discuss the above questions, along with the limitations of our study, in this section.

### 5.1 Snow Persistence: A New Descriptor of Catchment Hydrologic Response

Our results suggest that SP could explain shape-based streamflow signatures related to low flows and, to a lesser extent, high flows. In terms of stream low flow signatures, links between SP and baseflow index, between SP and normalized $Q_5$, and between SP and low flow variability (Low-FDC) exist across a wide range of aridity levels and climatic seasonalities, across Canada and the United States. The strength of the associations between SP and shape-based stream low flow signatures, however, varies with climatic aridities and seasonalities. For example, SP could more strongly explain BFI in dry and in-phase climates compared to wet and out-of-phase climates. These results imply that SP could be used as a new catchment climatic descriptor in explaining stream low flow shape-based signatures across a vast range of climatic conditions. Furthermore, there are associations between SP and stream high flow shape-based signatures (normalized $Q_{95}$ and High-FDC) in catchments with out-of-phase precipitation seasonality and moderately wet to very arid climates. These high flow results are supported by previous research that suggested that there are important links between SP and the timing and magnitude of peak flows, wherein snowmelt is a primary source of high flow generation in catchments with out-of-phase (winter-dominated) precipitation seasonality (Kampf and Lefsky, 2016).

These findings have implications for developing parsimonious large-scale models to predict streamflow behaviour (particularly low flow conditions). Indeed, a simple metric of snow accumulation and melt, such as SP, can be used to improve the performance of large-sample hydrology methodologies for hydrologic knowledge generalization and extrapolation to ungauged basins. SP can be a new candidate for further discovery and advancements in the field of large-sample hydrology and predictions in ungauged basins around the world with diverse climates, land cover, topographies, and geologies, particularly in conjunction with more climatic and physically based catchment variables (e.g., topographic, geologic, vegetation).



### 5.2. How Does Snow Persistence Control Streamflow Shape-based Signatures?

Our findings suggest that a longer persistence of snow on the ground could preserve stream baseflow, increase the magnitude of
extreme low flow relative to average flow, and reduce low flow variability (Figs. 3a-c; respectively). Previous studies have also
suggested that there is a notable relationship between snow and increased stream baseflows (Rumsey et al., 2015; Jenicek et al.,
2016; Godsey et al., 2014). Snow persistence is a simple integrated metric that reveals the accumulation and melt patterns of
snowpack (Hammond et al., 2018b) and could control stream low flow dependencies on subsurface water storage (Siirila-
Woodburn et al., 2021; Brooks et al., 2021). Indeed, snowmelt could feed subsurface water storage (Wu et al., 2020; Hayashi,
2020; Hammond et al., 2019; Pavlovskii et al., 2018). In turn, a larger subsurface water storage could lead to a larger (and more
stable over time) subsurface-stream interaction (Bloomfield et al., 2009; Tashie et al., 2020), during periods of no or low
precipitation.

Our results further show that snow persistence becomes a stronger factor in controlling the magnitude of stream low flow relative
to stormflow (or average flow), in regions with a relatively high aridity as well as high synchronicity between temperature and
precipitation (e.g., the central United States & Canada), as reflected in BFI and normalized $Q_5$ dependencies on snow persistence,
aridity and precipitation seasonality (Figs. 3a & 3b). Indeed, high aridity and large synchronicity between summer rainfall and
high summer temperature could lead to a lower rainfall contribution to subsurface water storage, making snow-based water storages
from spring snowmelt a major control on the magnitude of subsurface water storage and, ultimately, stream low flow (Fig. 4a). In
wet/out-of-phase (or winter dominated) climates, such as along the Pacific and Atlantic Maritimes in Canada or the coastal
mountain ranges in the northwestern United States, lower aridities and a larger contribution of precipitation desynchronized with
temperature (e.g., fall and early winter rainfall) could reduce the relative contribution of winter snowfall (or spring snowmelt) to
subsurface water storage and ultimately low flow (Fig. 4b). This is reflected in lower effects of snow persistence on BFI and
normalized $Q_5$ in wet out-of-phase climate compared to dry in-phase climate (Figs. 3a & 3b) following the hypotheses of Hammond
et al. (2018a).

The findings also suggest that snow persistence influences stream high flow signatures along a narrow range of aridity and
seasonality (Figs. 3d & 3e). A larger snow persistence is associated with larger variability in stream high flows and a larger ratio
of high flow relative to average flow (i.e. flashier stream high flow) in arid out-of-phase climates (e.g., Nevada and Utah). In arid
climates with winter-dominated (out-of-phase) precipitation, rapid snowmelt could be one of the major sources of stream high flow
generation during short melting periods (e.g., late spring or early summer). Stein et al. (2021) and Janssen et al. (2021) classified
mountain areas of Nevada and Utah as snow-dominated, and Stein et al. (2021) suggested that snowmelt and short summer rainfalls
are the major processes controlling flood generation across these two areas. We hypothesize that large snowmelt in a short period
of time, due to a rapid increase in temperature in early summer, could create a relatively large pulse of water moving laterally over
the surface or through the shallow subsurface toward streams, leading to a large variability in high flow and its markedly higher
magnitude compared to average flow (Fig. 4c). In other regions with in-phase seasonalities (summer-dominated climate) and/or
wet climate, there are less strong (or no) associations between snow persistence and stream high flow shape-based signatures (Fig.
4d). Excess or sustained rainfall and rain-on-snow could be the major processes controlling high streamflow in these regions (Stein
et al., 2021). More detailed research on how and why snowpack accumulation and melt could (or could not) be the factors
controlling stream high flow generation could further illuminate links between SP and high flows.





Overall, the statistical models generated in this study could be used to generalize low flow and high flow characteristics across vast ungauged areas and/or a under changing climate, as will be discussed in next sections. Our results show that the differences in seasonality and aridity of catchments can cause differentiated associations between SP and streamflow signatures (e.g., high & 360 low flow generation) across North America. Thus, future research could explore such differentiations in-depth using mechanistic models to gain a mechanistic understanding of these hydrological processes.

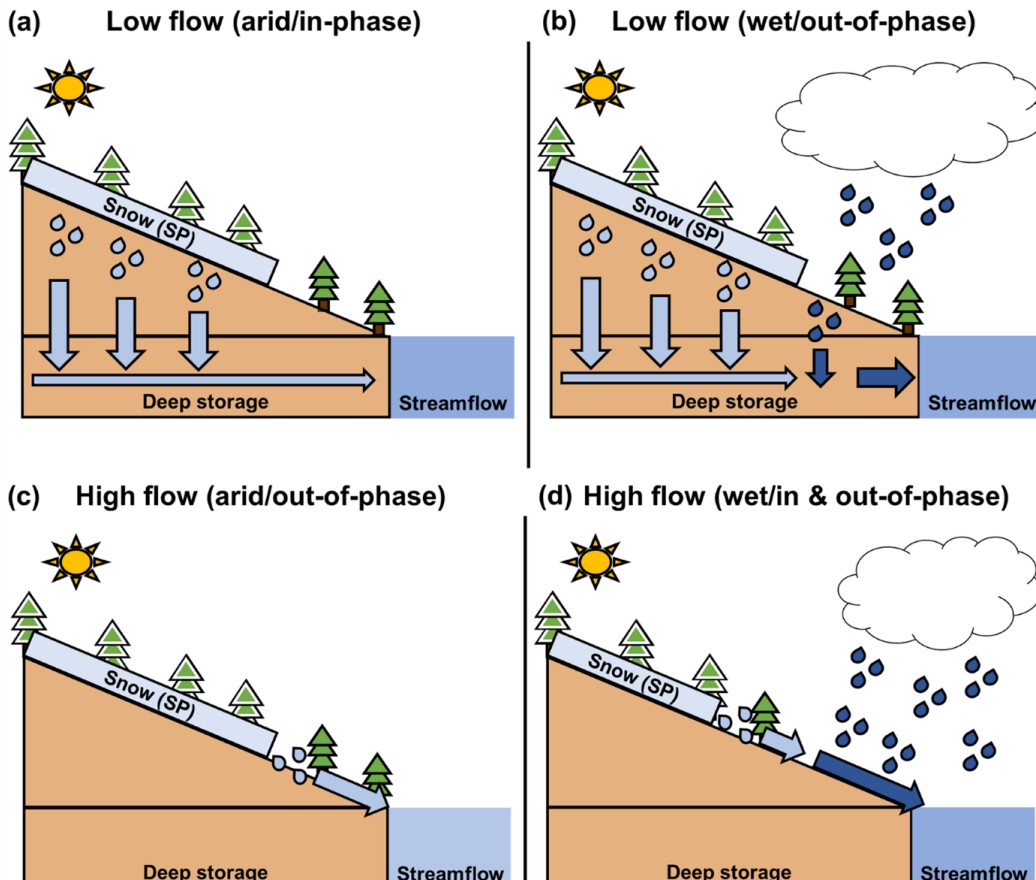

**Figure 4. Conceptualization of snow persistence effects on (a) low flow in dry/in-phase climate, (b) low flow in wet/out-of-phase climate,**
**(c) high flow in dry/out-of-phase climates, and (d) high flow in wet & dry/in-phase climates. These graphical illustrations represent potential mechanistic links between this study's obtained associations between snow persistence (SP) and streamflow signatures. Light blue represents snowmelt contributions to streamflow; moderate blue indicates mixtures of rain/snow contributions; dark blue represents rainfall-based contributions to streamflow. In (a), within dry and in-phase climates, the stream low flow could be strongly influenced by snow accumulation and melt patterns. High aridity and large synchronicity between summer rainfall and high summer**
**temperature could lead to a lower rainfall contribution to subsurface water storage, making snow-based water storages from spring snowmelt a major control on the magnitude of subsurface water storage. In (b), within wet/out-of-phase climates, the links between snow persistence and stream low flow could be additionally influenced by other water sources (such as rainfall), leading to a relatively lesser impact of snow accumulation and melt patterns on stream low flow. In (c), the relationships between snow persistence and high flow in arid/out-of-phase climates could be influenced by relatively rapid early summer melt, leading to a large pulse of snowmelt generated**
**stream high flow; whereas in (d) such snow-based effects in wet (in & out-of-phase climates) are potentially outweighed by other sources of high flow such as rapid intense rainfall or rain on snow.**





### 5.3 Implications for Climate Change-Induced Effects on Catchments

Catchments in North America (and around the world) are highly vulnerable to snow loss from climate change (Huning and Aghakouchak, 2018; Shrestha et al., 2021; IPCC, 2021; Siirila-Woodburn et al., 2021). Recent research studies suggested that

within the next 35-60 years, low-to-no-snow conditions resulting from unabated climate change will become persistent across large parts of western North America (Siirila-Woodburn et al., 2021). Furthermore, the timing of snowmelt on an annual level is expected to be earlier in the year (Musselman et al., 2021). All this paints a picture of a future with less persistence of snow on the ground if current warming continues. The growing aridification of large parts of North America (Overpeck and Udall, 2020) and potential changes in the annual precipitation timing (or seasonality) of large regions of the continent (Gershunov et al., 2019; Easterling et

al., 2017; Zhang et al., 2019), along with the aforementioned changes in snow presence on the ground, emphasize the need for a regionally-sensitive snow-related metric to identify when and where alterations in snowmelt timing as well as snowpack loss are poignantly detrimental to the hydrology of catchment systems (Siirila-Woodburn et al., 2021). Our paper takes a step in this regard by identifying under which seasonality and aridity ranges, the (changes in) persistence of snow on the ground influences streamflow characteristics at low and high flows.


In the context of the studied streamflow signatures, the results show that decreases in snow persistence could result in lowered baseflows and a larger variability in low streamflow across many catchments in North America. In moderately wet – extremely dry and out-of-phase seasonality, there may be lower hydrograph peaks (relative to average flow) and a lesser high flow variability. Additionally, the fact that there is likely to be a potential large-scale aridification in North America (Overpeck and Udall, 2020)

highlights the usefulness of snow persistence metric in helping to describe hydrologic behaviour at low and high flow conditions in a more arid future climate. This study's analysis shows that as aridity increases, the effect of snow persistence becomes stronger on sustaining baseflow, reducing hydrograph low flow flashiness and raising hydrograph peaks. Future research can focus on the classification and integration of regional changes in aridity and seasonality to snow persistence-related work.

Future work can also use snow persistence, along with ecologically significant metrics of catchment biological viability (McMillan, 2020), to inform catchment management efforts (Siirila-Woodburn et al., 2021) by creating thresholds, models, or even practical guidelines for levels of snow persistence required to sustain catchment ecosystems and anthropogenic activities. This can provide direct and actionable advice and insights for practitioners who are responsible for managing or using such natural resources. Another area of fruitful research could explore the relationships between snow persistence and hydrologic behaviour in response

to higher-order impacts of climate change. For example, greater wildfire risk or tree mortality in many catchments in North America (Westerling et al., 2006; Abatzoglou and Williams, 2016; Whitman et al., 2015) are likely to have effects on forest snow ablation (Gleason et al., 2013) and snow persistence and ultimately on streamflow characteristics (Siirila-Woodburn et al., 2021). Such interplays and feedback could prove interesting in advancing knowledge, understanding, and importantly predictability/forcasting of climate change effects on hydrology.

### 5.4 Limitations and Uncertainties of the Study

As we expected, the performance measure (i.e. $R^2$) of our statistical models were not very high, mainly because of the absence of other factors that we didn't consider in the analyses, such as geology and topography (and/or their associated interactions with climate). We know these factors play key roles in explaining shape-based signatures at stream high/low flows. The importance of these factors were confirmed in both large-sample hydrology studies (Addor et al., 2018; Janssen and Ameli, 2021; Sawicz et al.,

2011; McMillan, 2020; Kuentz et al., 2017) and in theoretical and field-based catchment-scale analyses (Carlier et al., 2019; Jencso



and McGlynn, 2011; Müller et al., 2014; Pfister et al., 2017). Their importants has also been shown in previous studies at the hillslope-scale (Ameli et al., 2015; Ameli et al., 2018; Hopp and Mcdonnell, 2009). We did not include these attributes (and their interactions with climatic metrics) because our goal was not to get the most accurate model, but instead to identify how far we can go in explaining detailed streamflow characteristics with one simple, widely available, and accurate satellite-based snow-related

metric. With current (and upcoming) advances in the development of widely-accessible and accurate data on catchment geology (Smelror, 2020), and new topographical data releases (Amatulli et al., 2018), future work could include snow persistence, among other explanatory factors of stream low/high flow signatures, to more robustly evaluate where, when, and why snowpack changes impact streamflow hydrograph shape.

Echoing previous work on the snow persistence product, it is known that cloud cover can disrupt snow detection (especially in winter months) for daily snowfall data of up to 22-50% for discontinuous snow cover (Molotch and Meromy, 2014). Similarly, forest cover can disrupt snow detection in densely forested regions (Rittger et al., 2013). To reduce these impacts, this study used a newer version of remotely sensed data that attempts to better algorithmically correct snowfall detection errors (Riggs et al., 2016). Further research and corrections may be required to create a higher quality global SP product; particularly for regions such as

Oceania, Europe, or South America (Hammond et al., 2018b). Such improved data could be used to evaluate the association between snow persistence and streamflow signatures at the global scale across a larger spectrum of aridity and seasonality than considered in our paper. This is especially important given this study's findings on the potential outsized impact small amounts of snow can have on catchment behaviour, particularly in arid climates.

## 6. Conclusion

Snow persistence is a globally available, remotely sensed measure of how long snow exists on the ground within a given catchment. There is emerging evidence that it is a powerful descriptor of catchment behaviour, but such work has previously focused on a smaller range of catchment behaviours and climatic conditions. This study sought to expand such work towards explaining hydrograph variability (shape) in low and high flow conditions across a broad range of North American climatic regimes. Through statistical analyses, it has been shown that there are generally statistically significant associations between snow persistence and

stream low flow behaviour in the form of decreasing low flow variability, sustaining catchment baseflows, and increasing stream low flows relative to average flows, regardless of the magnitude of climatic aridity and precipitation seasonality. The findings also depict the associations between snow persistence and increased high flow magnitude relative to average flows, as well as increased high flow variability for catchments in moderately wet to dry climates with winter-dominated (out-of-phase) precipitation seasonalities. These findings are particularly relevant for practitioners to better understand the interplays and feedback between

climatic variables and hydrology in the light of climate change in the years to come. Future work can be done to expand the range of descriptors used within hydrological studies to include snow persistence as a climatic metric, and to mechanistically explore the relationships between snow accumulation, melt and hydrologic behaviour, and further develop the fundamental knowledge that supports global scale snow science and hydrology.

## 7. Code Availability

The code for statistical analyses, visualization and Johnson-Neyman analysis will be published on a GitHub repository upon the publication of the paper.



**8. Data Availability**

United States snow persistence data is available at: https://doi.org/10.5066/P9U7U5FP. Canada snow persistence data is available at: https://doi.org/10.4211/hs.82326ee6288241f3a85ed1d01a3083d0. U.S. and Canada streamflow data are publicly available from

Environment and Climate Change Canada and USGS. The climatic and streamflow data processed for each catchment will be published on a GitHub repository upon the publication of the paper.

**9. Team List**

- Edward Le (EL)
- Ali A. Ameli (AAA)
- Joseph Janssen (JJ)
- John Hammond (JH)

**10. Author Contributions**

EL: Data Curation, Investigation, Software, Visualization, Writing (Draft Preparation, Editing, and Review), Formal Analysis

AAA: Conceptualization, Funding Acquisition, Methodology, Project Administration, Resources, Supervision, Data Curation,
Writing (Editing, and Review)

JJ: Formal Analysis, Investigation, Software, Validation, Writing (Editing and Review)

JH: Data Curation, Software, Writing (Editing and Review)

**11. Competing Interests**

The authors declare that they have no conflict of interest.

**12. Acknowledgements**

This research was funded by the NSERC-Funded Canadian Statistical Science Institute (CANSSI) grant awarded to AAA. Kim Janzen, at the University of Saskatchewan, is thanked for assistance with the final manuscript editing. We also acknowledge the efforts of Kristo Elijah Krugger in providing the initial code for the Johnson-Neyman analysis of the data.

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
