# Peer review of "Now at U.S. Geological Survey, MD-DE-DC Water Science Center, Baltimo"

_Hydrology and Earth System Sciences, 2022_

## Author Comment (AC1)

**"Snow Persistence Explains Stream High Flow and Low Flow Signatures with Differing Relationships by Aridity and Climatic Seasonality" by Le et al.**

**Response to Anonymous Referee #1 by Le et al.**

**Referee #1 comments:**

Authors analysed how snow persistence, aridity, and seasonality conditions control streamflow, specifically flashiness at low and high flows in 1187 catchments across United States and Canada. They addressed two main research questions, which are 1) Can snow persistence explain the shape of streamflow hydrographs at low and high flow conditions? and 2) How do aridity and seasonality affect the ability of snow persistence to explaining shape-based signatures at stream low and high flow conditions? Authors found that for low flow conditions, larger snow persistence increases baseflow and reduces low flow variability regardless catchment aridity and seasonality. Authors also showed that snow persistence became a stronger factor in controlling baseflow, in regions with a relatively high aridity and/or with summer-dominant precipitation regimes. Based on analyses, authors concluded that snow persistence may serve as a useful streamflow descriptor across different climates and runoff regimes.

In my opinion, authors did an interesting work. I agree with authors, that the use of snow persistence as a metric describing catchment response across a wide range of climates is novel. Besides that, I see the novelty that this metric can be derived from satellite data and thus it can be used at a global scale. I also like that author related the snow persistence to different aridity and seasonality of catchments. Although the results are not much surprising as they mostly confirm our existing knowledge, I found the study important and novel, thus appropriate for HESS. However, I have several comments listed below, which should be addressed before I can recommend the manuscript for publication.

**Major comments**

After reading of abstract and study objectives, I was really motivated in further reading since I was curious about what authors investigated. However, I was a bit disappointed, because in my opinion, the study conclusions are not sufficiently supported by results and illustrations. In my opinion, only two figures showing study results (Fig. 2 and 3) are too few to draw general conclusions. Therefore, I would encourage authors to support their results with further analysis and figures. In comments below, I tried to make a few suggestions which authors may consider.

In my opinion, it might be interesting to look whether the snow persistence is a good predictor for the selected runoff signatures in years with snow-poor and snow-rich winters (or dry/wet, cold/warm years). Comparing statistics of individual years instead of mean statistics of the whole study period may allow for a more direct attribution of the inter-annual variations of snowpack to variations in runoff characteristics.

I see the evidence provided by authors that snow persistence can partly explain the selected streamflow signatures. However, the snow regime belongs to the main component of the water balance in highelevation and/or high-latitude catchments next to precipitation and its seasonality and evapotranspiration. Therefore, similar results would be maybe achieved for any of these characteristics. It means, that not only snow persistence might explain streamflow characteristics, but also aridity or seasonality indexes might bring similar results. Therefore, to further support existing results, it might be interesting to look how strong is the snow persistence as a predictor compared to aridity and seasonality indexes. Maybe, at least some correlation analysis comparing the predictive strength of all predictors might be beneficial. In addition to my above comments, results section should be extended. As it is now, it contains only a short description of results shown in Fig. 3 and it looks unproportionally short compared to the discussion section. As I mentioned above, interpretation based on one or two figures seems unconvincingly to me and I would encourage the authors to add more analysis and related interpretation which may further support (so far interesting) results.

**Specific comments**
L 105: I would somewhere mention the basic statistics of the study catchments (e.g., as a range of values), such as area, elevation, annual precipitation, snow persistence, etc.
L 110: I suggest including equations of how the main characteristics (snow persistence, aridity index, seasonality index) have been calculated.
L 147: Maybe I did not understand correctly, but 30% of area difference between the two different approaches of area calculation sounds as a large difference. Why is it so much?
L149: I understand that only perennial rivers were considered for the analysis. Nevertheless, would the results be different if also river intermittency would be considered? Please discuss shortly.
L173: $30^{th}$ and $70^{th}$ percentiles for low- and high FDCs sound rather as arbitrary choice. Is there any reason for choosing exactly those thresholds?
Authors defined several streamflow signatures for the analysis. This is fine, but I would suggest including a few more, for example low flow duration or deficit volumes. Especially the former might be beneficial to further explain the role of snow persistence on low flow regime.
L 195: How the last day with snow presence has been calculated? Due to elevation range of individual catchments, the snow may be melted at lower elevations while some snow may be still present at higher elevations. Please clarify shortly.
L286-287: The research questions here are repetition from above, please considerer whether it is needed to introduce them again at this place.
L320: Maybe I missed it somehow, but I do not see reducing low flow variability from Fig. 3a-c as noted by authors. Please, add some more explanation.
**Technical corrections**
L154-155: I would omit these two lines.
L296: "snowpack" rather than "snow pack".
Fig. 4: Consider adjusting light blue and moderate blue colours in the figure since one can hardly see the difference between both.
L388: Perhaps, the brackets in "changes in" are not necessary.
L606: If I checked correctly, Muñoz-Sabater et al. (2021) has already the final paper published.

**Response to Referee #1 Comments:**

**Response to Major Comments**

We thank the reviewer for their helpful and positive comments.

In response to the major comments regarding the analyses, in our next revision of the manuscript, we will expand the set of analyses. First, as suggested by the reviewer, we will include additional analyses of high and low flow behaviour through the inclusion of high flow duration and low flow duration signatures. Our preliminary results show a strong relationship between snow persistence and both low and high flow durations, following the same directions as estimated for other low flow and high flow signatures used in the original manuscript.

Next, we believe that the suggestion to compare individual year statistics is an interesting direction to take this paper and we will explore correlational analyses of high snow persistent/low snow persistent years as well as across dry and wet years. We will then compare the correlational effects of aridity, seasonality, and snow persistence to further improve our analysis of the relationships between snow persistence and runoff behaviour in various climatic conditions. Through these additional analyses, we aim to provide a more direct attribution of the variations in snowpack to variations in runoff characteristics.

**Response to Specific Comments**

In response to the specific comments proposed by the reviewer, we will further clarify our study's methodology to help address these comments. We will also extract one additional figure from Fig. 3 to further illustrate and explain the interactive relationships between snow persistence, aridity index and seasonality index. This figure would show the regression lines between snow persistence and each signature at different levels (1st quartile, median, and 3rd quartile) of the aridity index and seasonality index. We believe this figure would clearly illustrate the impact of snow persistence on our shape-based streamflow signatures, which might be hard for readers to visualize from Fig. 3 alone.

**Response to Technical corrections**

We agree with the suggested technical corrections and will fix them in the revised manuscript.

---

## Author Comment (AC2)

**"Snow Persistence Explains Stream High Flow and Low Flow Signatures with Differing Relationships by Aridity and Climatic Seasonality" by Le et al.**

**Response to Anonymous Referee #2 by Le et al.**

**Referee #2 comments:**

**1. General comments**

Le et al. analysed how well snow persistence (along with aridity and precipitation seasonality) can explain a range of flow signatures in over 1000 catchments in North America, using a 19-year data set. They applied a linear model with interaction term (multilinear regression) and visually analysed the influence of snow persistence on each response variable. With a very short results section and basically only one figure (Figure 3), they come to significant results, such as that snow persistence influences low flow characteristics, and in some climate regions also high flow characteristics. Furthermore, the authors established a link between the spatial changes observed and future climatic changes, such as how a reduction in snow persistence could change flow characteristics.

In my opinion, the fitted linear models are not able to capture the variance sufficiently to draw these essential conclusions. The authors report values for  $R^2$  ranging from 0.11 to 0.25 (Table 1). Similarly (or as a consequence), the explained effect of snow persistence on the response variables is small: the largest values on the y-axes in Fig. 3 cover only 0.1% (for Q95) to 4% (for Q5) of the indicated interquartile range in Table 1. The effect is statistically significant, as mentioned by the authors, but in my view too small to be relevant. This is a common problem: as sample size increases, decreasing effects become statistically significant. The authors need to find ways to create models with greater predictive value that are able to produce effects of relevant size. In my opinion, the small effect size of the linear models makes this manuscript too weak to be considered for publication in HESS. I will explain this in more detail in the next section.

**2 Specific comments on the small effect size**

The authors discuss these low  $R^2$  values in their "Limitations" section and mention that this is to be expected as geological and topographical factors were not included. They cite Addor et al. (2018), for example, who considered these factors important. I disagree with this expectation of low  $R^2$  values and also with the explanation: Addor et al. (2018), a very similar but much more comprehensive study (barely cited by Le et al.), concluded "... that climatic attributes are by far the most influential predictors for signatures that can be well predicted based on catchment attributes". Instead of simple linear models, they trained Random Forests and found that they could explain large parts of the variances of signatures such as  $Q_{95}$  ( $R^2 > 0.8$ ),  $Q_5$  ( $R^2 \sim 0.6$ ) and BFI ( $R^2 \sim 0.5$ ) with climatic attributes alone (read from their Figure 5). These values are much larger compared to those reported here. Only for the slope of the flow duration curve was a similarly small  $R^2$  value reported.

The reasons for the larger  $R^2$  values reported by Addor et al. (2018) could be that they used

- more and other climatic variables
- more complex models
- a longer dataset, limited to the US.

The first item is important for the aim of the Le et al. manuscript, namely to show the predictive value of snowpack persistence (SP). As the authors indicate, snowpack persistence is easier to determine compared to snowfall fraction (which was used by Addor et al., 2008) and is therefore a very interesting and globally available predictor variable. To show the predictive value of SP, I would suggest repeating the Addor et al. (2018) study for the US and Canadian datasets and only use their climatic variables, then replace the snowfall fraction with SP and then remove step-wise all other climatic variables until the three used here remain (i.e. SP, seasonality of precipitation and aridity). With this setup, one can find out what the authors were aiming for, namely (line 418ff): "how far we can go in explaining detailed streamflow characteristics with a simple, widely available and accurate satellite-based snow-related metric" (along with seasonality and aridity).

**Response to Referee #2 Comments:**

We thank the reviewer for taking time reviewing our paper. Please see below 1) our response to the reviewer's comment on effect size, 2-4) our response to the reviewer's comments on the use of an alternative method with a better predictive performance in order to obtain a larger  $\mathbb{R}^2$ , 5) scientific importance of the effects sizes of our models, 6) detailed explanation of the use of random forest to estimate shape-based streamflow signatures used in our paper, and 7) the modifications we are willing to make in the revised manuscript.

**1) Response to reviewer's comments on small effect size and p-value:**

The reviewer argued that the significant p-values we obtained are mostly due to the large sample size of our study. We agree with the reviewer that blindly following conclusions based on meeting an arbitrary p-value threshold (usually 0.05) can lead to poor scientific conclusions. In part we fight against this in our paper by instead considering a much stricter p-value threshold of 0.01 while also correcting for multiple comparisons. Furthermore, the reviewer argued that the effect sizes of our regression models are small, and therefore they are not relevant to make conclusions. We disagree with the reviewer in this regard and we argue that our conclusions based on the effect size are relevant, and we focus on this for the remainder of response # 1. As recently explained by Anderson, Slater, Dadson, Blum, and Prosdocimi (2022), in a methodologically similar paper as our paper, a small effect size can be very important ("...a 1%-point increase in catchment urban area results in a small (0.6%–0.7%), but highly significant increase....." see the abstract of the paper). Note that the effect sizes in Figure 3 of our paper are referring to impacts on transformed streamflow signatures and once we consider the impacts on untransformed signatures these effect sizes are larger for most of signatures than what is shown in Figure 3. More importantly, the decision on whether the effect size is relevant or not should consider the possible ranges of the covariates. As will be explained in detail below (response # 5), our results show that a probable change in snow persistence could decrease BFI from 0.62 to 0.55, could increase the slope of flow duration curve at low flow condition from 2.34 to 3.67 (57% increase in the slope of flow duration curve at low flow condition), and could pronouncedly decrease low flow event duration in each year from 57 days to 25 days. So, the effect sizes of our model show a large influence of snow persistence on low flow stability and duration, suggesting large impacts of climate change on low flow variation and stability. Also, we have to clarify that interquartile ranges in Table 1 represent the range of untransformed attributes/signatures, but Figure 3 (in the original under review paper) shows the impacts on transformed signatures.

**2) Response to reviewer's comment on the use of random forests**

The reviewer argued that our  $R^2$  values are small and we could have obtained a much larger  $R^2$  as done in Addor et al. (2018) using random forests. First, we have to clarify that we *did not* evaluate Q5 and Q95 in our paper. Q5 and Q95 are magnitude-based signatures and are easily predictable using climatic attributes as shown in Addor et al. (2018) for the catchments across the United States. Instead, we evaluated Normalized Q5 and Normalized Q95 in our paper, which are shapebased signatures and reflect the functionality of catchments and refer to flashiness of streamflow hydrographs. These signatures are hard to predict using climatic attributes alone and refer to the ratio between Q5 (or Q95) and average flow. They depend on the presence of macro-pores and soil and bedrock properties and other catchment internal processes controlling streamflow generation during low-flow and high-flow. Please refer to Janssen and Ameli (2021) and McMillan (2021) for details on the differences between magnitude-based versus shape-based signatures and different mechanisms (and attributes) controlling these signatures. What the reviewer suggested (i.e. the use of random forest) was already done in our previous paper (Janssen & Ameli, 2021) and we obtained small cross validation  $R^2s$  for Normalized Q5 and Normalized Q95 with climatic attributes (including snow fraction) and using random forests (see point # 6). Indeed, the  $R^2$  values with random forest would be only slightly larger than what we obtained here using our simple 6-term statistical model (21% versus 12% for normalized Q5 & 24% versus 11% for normalized Q95 & 27% versus 23% for Base Flow Index). We have to clarify that the range of our  $R^2$  for different shape-based streamflow signatures will be between 11% to 51% in the revised manuscript, after adding two signatures (i.e. Low Flow Event Duration & High Flow Event Duration) that the other reviewer suggested.

**3) Response to reviewer's comment on small R2**

The reviewer suggested that the authors need to find ways to create models with greater *predictive* performance in order to show the *predictive* value of snow persistence. However, in this manuscript, our goal is inference and not prediction. See Efron (2020) and Shmueli (2010) for detailed discussions on the differences between the two statistical goals of estimation (or explanation or inference or description) versus prediction. As Shmueli (2010) clarified, the correct model that reflects the data generating process may actually have worse predictions compared to a strongly predictive model. When the goal is inference, unbiased models are preferred, but random forests increase predictive capability compared to linear regression by increasing bias while decreasing variance. Again, referring to Anderson et al. (2022), in a methodologically similar paper as our paper, they did not even show  $R^2$  or any other performance measure of their model. Paying less attention to model's fit in inference (or explanation) studies is fairly routine when investigating hydrological behaviour (and in other disciplines), and researchers frequently publish high impact papers in high impact journals with  $R^2$  smaller than 0.05. Again, we emphasize that the goal of this study is inference, not prediction. In other words, our goal is to quantify the functional relationship between the covariate (snow persistence) and response (streamflow signatures), and not to quantify the proportion of the response variance explained by the covariates (which is what the  $R^2$  value measures). To further illustrate this difference, we consider a simple simulation of two datasets (see Figure 1 below): both datasets are generated from the same linear model but have different error variances. In Figure 1, for both cases, we plot the data, the true regression slope and estimated regression slope (inferred from data) and its confidence intervals. In both cases, there is a small but non-zero (and statically significant) effect which we can correctly estimate, but the R2 values are 0.81 vs. 0.01. Small R2 only shows that there is unexplained variance in the response. The p-value of the functional relationship tells us how much evidence we are seeing in our data for a significant functional relationship. The figure clarifies that a model with very poor predictions could lead to an accurate (and statistically significant) estimation of a functional relationship between a covariate and response. The unexplained variance could be due to other covariates not included in the model or could be purely random noise. We already knew that our shape-based signatures are hard to predict using climatic data alone based on our random forest analyses conducted in Janssen and Ameli (2021) (see responses # 2 & 6 for more details). These signatures are dependent on the presence of macro-pores and soil and bedrock properties and other catchment internal processes. We only have highly uncertain data about these attributes and processes even in extensively studied regions. These points were clarified in the discussion section of the under-review paper and will be further clarified in the revised version.

The goal of our paper was to explore the functional relationship between globally available snow persistence data (Hammond, Saavedra, & Kampf, 2018) and shape-based signatures at different levels of aridity and seasonality, and not quantifying the predictive power of snow persistence. Our results show that for some signatures, snow persistence is strongly related to signatures and for some others snow persistence is moderately related to signatures (see response #1), regardless of R2 values. More importantly, our results emphasized that this widely available data (i.e. snow persistence) can be used for inferring the climate change impacts on shape-based signatures across the globe, as climate change strongly impacts the timing of snow presence on the ground. There might be other climatic attributes, available in some regions, with a larger predictive power than snow persistence. However, the advantage of snow persistence is that it is widely available through satellite data, and our results further showed that it can provide direct insight about the impact of snow presence/loss on streamflow hydrograph shape/flashiness. Therefore, despite we could not obtain a much larger R2 using more complicated machine learning model as explained in response # 2, we believe that a larger or smaller  $R^2$  does not impact our inferences and interpretations in the way that we formulated and stated such inferences in our paper (and we will further clarify this in the revised version).

Figure 1: Illustration of the estimation of true beta value (i.e. functional relationship between covariate and response) using simple linear regression. Here we consider a simple simulation of two datasets, where both datasets are generated from the same linear model but have different error variances. In both cases, there is a small but non-zero effect which we correctly estimate but the  $R^2$  values are 0.81 vs. 0.01. This figure clarifies that a model with very poor predictions could lead to an accurate estimation of a functional relationship between a covariate and response. The code for this simulation study can be found at <a href="https://github.com/hgwm">https://github.com/hgwm</a>.

**4) Response to reviewer's comment on the use of an alternative method**

We believe that our statistical analysis adequately addresses the objectives of our paper. We designed a statistical experiment based on background knowledge and based on the objective to be explored. Our objective is: investigate how snow persistence interacts with climatic aridity and seasonality to impact shape-based signatures. This objective is clear throughout the paper including in the title, abstract, introduction, discussion and conclusion. A complicated random

forest model with many, somewhat arbitrarily chosen, parameters and terms cannot explore our objectives as explained in detail in the discussion section of Janssen and Ameli (2021) (see section 5.2 of the cited paper). Generally, advanced machine learning models generate several intermediate functions which may not be interpretable and scientifically supported. Janssen and Ameli (2021) opened the black box of random forest used for the prediction of shape-based signatures (the same signatures used in the under-review paper) and explained that there is no scientific support for several of the interaction terms that random forests identified. Here, we are using a simple 6-term model. Despite its simplicity, the model was designed to sufficiently and directly incorporate and test the objectives of the paper i.e. the interactive behaviours amongst snow persistence, climatic aridity and climatic seasonality. Paraphrasing Einstein, "the maxim is: make a model as simple as possible, but not simpler than that" (Savenije, 2010). It is the philosophy that we followed in design of our statistical models. We have to acknowledge that this research was funded by the grant awarded by Canadian Statistical Sciences Institute. We believe that the design of the statistical experiments in our paper leveraged the state-of-the-art techniques in statistical inference and we selected the most appropriate statistical and visualization tools to explore our objectives.

**5) Evaluating the scientific importance of our models' effects sizes (details & examples)**

Here, we build on our statistical model results, and evaluate and explain how much a probable change in snow persistence can alter four shape-based streamflow signatures used in our paper (the same interpretation can be conducted on the rest of the signatures used in our paper as we will conduct in the revised manuscript). Multiple studies have reported an earlier snowmelt timing across Canada and the U.S. over the past several decades, and have projected earlier snowmelt for future periods (Clow, 2010; Hodgkins & Dudley, 2006; Musselman, Clark, Liu, Ikeda, & Rasmussen, 2017; Semmens & Ramage, 2013; Stewart, Cayan, & Dettinger, 2005). More recently, Broadbent et al. (2021) showed that snowmelt is predicted to occur 50-130 days earlier in alpine climate regions due to climate change by the end of the century. Or as Harpold and Brooks (2018) showed in Colorado, snow is already melting as much as a month earlier than the historical norm. Now let's consider a 60-day decline in snow presence on the ground in an Alpine / sub-Alpine region of North America (e.g., across the Rocky Mountains) in the future. In this region, Fig. 2 (in the original under review paper) shows a current long-term average of 75% snow persistence, seasonality index of ~ 0.09, and aridity index of ~ 0.72 (Ln(AI) = -0.32). The 60-day decline in snow presence on ground would imply a change in snow persistence from 75% to 42%. In the remainder of this section, we build on our statistical results and explore how much a change in snow persistence from 75% to 42% (with a fixed seasonality index of 0.09 & Ln(AI) = -0.32) alters a) BFI, b) Low-FDC, c) Low Flow Event Duration, d) Normalized Q5. Note that other regions with a more seasonal climate (e.g., SI=0.29) than what we consider here in this example, could show larger effects sizes of snow persistence on streamflow signatures based on our statistical results. So, the below examples, do not reflect the interpretation of our results for a region where our model shows the largest effects sizes of snow persistence on streamflow signatures.

**I. BFI**

The upper-middle panel of Figure 2 below shows that a potential change in snow persistence from 75% to 42% (60 days decrease in the presence of snow on the ground) could decline BFI from 0.62 to 0.55. For moderate to large catchments across the Rocky Mountains this implies a large decrease in the volume of available water in late spring and early summer in the future.